# Comparison of Metrics for Shape Quality Evaluation of Textures Produced by Laser Structuring by Remelting (Waveshape)

**DOI:** 10.3390/mi13040618

**Published:** 2022-04-14

**Authors:** Oleg Oreshkin, Alexander Platonov, Daniil Panov, Victor Petrovskiy

**Affiliations:** 1Moscow Engineering Physics Institute, National Research Nuclear University MEPhI, Kashirskoe Shosse 31, 115409 Moscow, Russia; zzolegoreshkin@rambler.ru (A.P.); zzzzzolegoreshkin@rambler.ru (V.P.); 2Center for Design, Manufacturing & Materials, Skolkovo Institute of Science and Technology, Bolshoy Boulevard 30, Bld. 1, 121205 Moscow, Russia; zzzzolegoreshkin@rambler.ru

**Keywords:** laser structuring, laser texturing, spectral analysis, spectral entropy, cross-correlation, texture shape, texture assessment, binary classification, Waveshape

## Abstract

The study is focused on investigating approaches for assessing the texture shape deviation obtained by laser structuring by remelting (Waveshape). A number of metrics such as Fourier spectrum harmonic ratio, cross-correlation coefficient (reverse value), and spectral entropy are investigated in terms of surface-texture shape deviation estimation. The metrics are compared with each other by testing two hypotheses: determination of target-like shape of texture (closest to harmonic shape) and determination of texture presence on the cross-section. Spectral entropy has the best statistical indicators for both hypotheses (Matthews correlation coefficient is equal to 0.70 and 0.77, respectively). The reverse cross-correlation coefficient proved to be close in terms of statistical indicators (Matthews correlation coefficient is equal to 0.58 and 0.75 for the first and second hypothesis), but is able to estimate the shape similarity of regular texture independent on its type. The provided metrics of shape assessment are not limited to the texturing process, so the presented results can be used in a broad range of scientific fields.

## 1. Introduction

Texture as a geometric three-dimensional surface pattern is an essential part of an object, and its size and shape can vary over a wide range. Surface texture can strongly influence the functional and aesthetic properties of a part (tribological, optical, hydrophobic, etc.). In addition, texture can play a role in changing the fatigue and strength properties of the part due to the redistribution of stress concentrators on the surface [1].

Applications of regular surface textures could be met in various fields: in medicine by creating implant textures for improving osteosynthesis [2], in the aerospace industry by generation of icephobic metal surfaces [3,4], in die molds manufacturing by fast and robust creation of mold texture [5], and for various tribological purposes [6], etc.

Regular surface textures can be achieved by various methods: micromilling [7,8], electrical discharge machining [9,10], etching [6], and electron beam processing (Surfi-sculpt^®^) [11]. In addition, lasers are a powerful tool for producing the regular textures. Numerous laser technological methods are developed, such as laser surface texturing (LST) by laser ablation [12,13], laser-induced periodic surface structures (LIPSS) [14], direct laser interference patterning (DLIP), Waveshape [15]. The area of interest in this research is laser structuring by remelting (Waveshape), and the shape deviations of textures generated by this method are analyzed in the paper.

### 1.1. Waveshape Process

Laser structuring by remelting (Waveshape) is an innovative approach for creating texture on metal surfaces. The process is based on melting a thin surface layer with laser radiation (Figure 1). The texture on the surface corresponds to the amplitude modulation of laser power. To obtain the textures, the average laser power $P_{M}$ (in the order of several hundreds of watts) is modulated during the process with the laser power amplitude $P_{A}$ (in the order of several tens of watts) at a target spatial wavelength λ. In surface processing, the laser beam with a beam diameter $d_{L}$ moves over the surface at a scan speed $v_{scan}$ [16,17,18].

The formation of the texture takes place during solidification. The active principle of the Waveshape. The red arrow indicates the movement direction of the laser beam. consists of the variation of the melt pool volume and the dependent movement of the three-phase line [16]. The direction of solidification follows the curvature of the melt pool and texturing is therefore achieved [19].

The size of textures created by Waveshape can be adjusted across a broad range. The spatial range of textures can vary from a few tens of microns to a few tens of millimeters. The height range of textures is from several microns to several millimeters [18].

### 1.2. Classification of Textures and Determination of Texture Deviation

Surface textures can be categorized into several types (Figure 2). For example, stochastic textures with random distribution of geometry, and deterministic textures where the distribution of geometry is specified in a known way. Deterministic textures can be classified as regular and irregular depending on the spatial repeatability of the pattern.

The same texture can be both deterministic and stochastic depending on the spatial range, for example, stochastic in the range of small spatial wavelengths (roughness) and deterministic in the range of large spatial wavelengths (regular waviness) or vice versa. For functional surfaces, mainly regular deterministic textures are applied, since often only a deterministic texture performs a specific function.

Regardless of the texturing method and application of the texture, the main goal of the process is achieving a required accuracy. This means that the obtained texture should have the defined match in size, position and shape to the target texture. In fact, any surface texture parameter can be assigned to one of these three clusters (Figure 3).

Texture size determination is a well-standardized practice. ISO 4287 includes a lot of profile parameters that describe different amplitudes and spatial dimensions. An advantageous approach for Ra measurements of profile cross-sections in different spatial ranges was introduced in [20]. Three-dimensional texture dimension parameters are described in ISO 25178. Dimensional texture properties almost always reflect absolute texture values, regardless of comparison with the target geometry.

Deterministic textures can also be measured by position (e.g., phase shift for harmonic textures) and shape. Shape measurement in a broad context means measuring similarity of obtained texture compared to the target texture independent on its dimensions. The shape of a texture can be important for functional textures, since the texture function is often determined not only by the texture size but also by its shape. The texture shape depends on the processing parameters, and matching degree between the obtained texture and the target texture can be one of the processing quality metrics. Shape quality refers to the similarity of the obtained and target textures. In this paper, the focus is on measuring texture shape.

Analyzing the textures generated by the Waveshape process, three research questions arise:How to measure that the obtained texture matches to the target texture?How to compare the shape quality of two generated textures with different dimensional parameters?Is it possible to check that the target texture is present on the cross-section?

Therefore, the research question consists of a suitable method to quantify the textural texture deviation.

In previous Waveshape investigations, the primary parameter for texture characterization was profile height [16,17,18,21]. Texture accuracy and texture deviation has so far been only marginally investigated. Temmler [18] introduced an asymmetry metric for harmonic textures as the asymmetry angle θ:(1)$$\theta =arctan(\Delta \lambda /h_{max}),$$
where Δλ is the peak deviation from the symmetry position in the lateral direction, and $h_{max}$ is the height amplitude of the texture (peak). Oreshkin et al. [22] concluded that the asymmetry angle is insufficient to estimate the accuracy, since even symmetrical textures can exhibit a shape deviation from the target shape. Therefore, they investigated the textural accuracy of sinusoidal textures using discrete Fourier transform analysis and found that it is mainly affected by shape deviation at a wavelength equal to half the wavelength used for texturing. They introduced the ratio of the magnitudes of the second and first harmonic peaks $R_{2/1}$ in the Fourier spectrum as a quality metric for harmonic textures. $R_{2/1}$ and θ have restrictions in implementation for evaluating complex textures.

In this study, four different metrics were used to estimate the similarity of target and obtained textures. The efficiency of similarity detection for each metric has been evaluated on harmonic textures generated by Waveshape.

Despite the analysis of textures produced by only one method, there are no apparent limitations on the use of metrics from the way the texture is created. In particular, the textures produced by LIPSS and DLIP methods are predominantly periodic, and their shape deviations can also be analyzed. Limitations of use related to the nature of the texture are mentioned in the discussion section.

The application of these quality metrics will be useful in future research for process quality analysis. Shape quality as one of the evaluation factors can be applied to find a suitable “parameter window” without shape deviation. Other possible implementations of the research results are data markup and processing assessment in machine learning applications.

This study analyzed four different metrics for evaluating texture shape quality obtained by the Waveshape method. A method based on binary classification to differentiate between surfaces with higher and lower quality, as well as to determine presence of a texture on the profile, was used. A comparison of the used metrics according to different statistical indicators was also carried out.

## 2. Materials and Methods

### 2.1. Laser Surface Processing

For the shape deviation analysis, the data set in the form of texture profiles obtained by the Waveshape was used. The textures are supposed to be harmonical.

The processing has been performed using an ytterbium fiber laser LS-2 (IPG IRE-Polus, Fryasino, Russia) emitting with a central wavelength of λ = 1070 nm with a maximum laser power of 2 kW. The laser beam diameter can be continuously adjusted by beam defocusing in the range of $d_{L}$ = 250–1500 μm. A three-dimensional scanning system consisting of an industrial robot IRB 4600 (ABB, Zürich, Switzerland) and a laser scanner (IPG Photonics, Oxford, MS, USA) provided scanning on a 300 × 300 mm field at a velocity up to 1500 mm/s. A process chamber with an argon atmosphere has been used to prevent oxidation. The residual oxygen concentration was maintained at a level less than 100 ppm and has been controlled with an oxygen meter AKPM-1-01 (Alfa Bassens, Moscow, Russia). The process parameters used for this study are given in the Table 1.

Ti-6Al-4V alloy has been used as a basic material for laser texturing. The material has been pregrinded up to roughness Ra<1μm. Before texturing, all samples were remelted with a laser radiation (melt depth of approx. 50 μm) to homogenize the samples and evaporate any non-metallic inclusions in the remelted zone.

For this study, 270 different textures have been obtained. The profiles dimensions and quality varied dramatically. On some profiles, the texture is indistinguishable. The height of textures varied in range of several hundred microns. The spatial wavelength λ of the textures ranged from 10 to 50 mm. Each profile has at least two waves. The profiles have been measured using a contact contourograph MarSurf XC 20 (Mahr, Göttingen, Germany). The octagonal sample with textured single lines is shown in the Figure 4.

### 2.2. Methodology of Metrics Evaluation and Comparison

The surface texture can be represented as a dependence between height at a point h(x) and its coordinate *x*. The Waveshape mainly creates periodic textures or superposition of periodic textures.

Thus, the height distribution h(x) (obtained texture) can be considered as a superposition of the target texture and some deviation. The nature of this deviation lies in the dynamics of the melt pool during texturing.

In this study, we apply different approaches implemented to measure the quality of textures: spectral methods, entropy measurement, correlation analysis between target textures and obtained texture. The approaches applied to one-dimensional texture cross-sections.

#### 2.2.1. Ratios of Spectral Magnitude

The spectral analysis can be applied due to periodicity of the textures created by the Waveshape. Cutting an integer number of fundamental wavelengths from the surface profile is used for correct analysis. Oreshkin et al. [22] applied the magnitude ratio of the second and the first harmonics $R_{2/1}$ (Figure 5) as a metric for texture deviation from symmetric shape. In the framework of this study, the spectral analysis of textures is extended to additional metric: ratio of the the sum of magnitudes greater than the second harmonic to the first harmonic $R_{all/1}$ (Figure 5).

Before calculating the metric, the preprocessing of the cross section consists of the following steps:Extraction of an integer number of waves of the obtained texture;Obtained texture alignment (high-pass filtering to remove the sample form from the profile).

After preprocessing and Fourier transform, the discrete spectrum $S(f_{n})$ provides information about the texture spectral density. Due to harmonic shape of the target texture, a peak corresponding to the main spatial wavelength $f_{main}$ stands out on spectrum. Since any obtained texture has additional frequencies in the spectrum, the ratio of the higher frequencies (or harmonics) to the main frequency (or harmonic) can be a shape deviation metric. The lower the value of the metric, the closer the shape of the obtained texture is to the harmonic one. Two metrics are considered here:The magnitude ratio of the second and the first harmonics $R_{2/1}$:
$$R_{2/1}=\frac{S(f_{2})}{S(f_{main})},$$
where $S(f_{main})$—spectral magnitude on the main frequency (the first harmonic), $S(f_{2})$ spectral magnitude on the doubled frequency (the second harmonic), $f_{2}=2f_{main}$.The ratio of the the sum of magnitudes greater than the second harmonic to the first harmonic $R_{all/1}$:
$$R_{all/1}=\frac{\sum_{n=2}^{n=s/2}S\left(f_{n}\right)}{S(f_{main})},$$
where $f_{s/2}$—half of the sampling frequency of the profile.

The summation of the deviation frequencies starts with the second harmonic, which is due to the desire to avoid spectral leakage of windowed texture, and by the assumption that the low-frequency part of the spectrum relies not on the surface shape, but on the surface form and lies outside the area of research interest.

#### 2.2.2. Reverse Cross-Correlation Coefficient ($CCC_{rev}$)

The use of cross-correlation is a common practice for accuracy estimation in signal theory [23]. In this study, the Pearson cross-correlation coefficient is used to determine the shape deviation of the obtained texture relative to the target texture.

As in the previous metrics, the profile is preprocessed as follows before the metric is evaluated:Extraction of an integer number of waves of the obtained texture;Alignment of the obtained texture (high-pass filtering to remove the sample form from the profile);Normalization of the target and the obtained profiles: minimum value is 0, maximum value is 1.

The cross-correlation coefficient CCC is evaluated against the normalized target texture. The preprocessing uses wavelength recalculation of the target texture, since the spatial wavelength deviation is not a shape deviation, but a size deviation.

Cross-correlation coefficient is calculated as following:$$CCC=\frac{max(corr\left(P_{targ},P_{obt}\right))}{\sqrt{max\left(corr\left(P_{targ},P_{targ}\right)\right)\times max\left(corr\left(P_{obt},P_{obt}\right)\right)}},$$
where $P_{targ}$—the target profile; $P_{obt}$—the obtained profile, $max(corr\left(P_{targ},P_{obt}\right))$—maximum value of the cross-correlation function; $max(corr\left(P_{targ},P_{targ}\right))$—maximum value of the autocorrelation function of the target profile; $max(corr\left(P_{obt},P_{obt}\right))$—maximal value of the autocorrelation function of the obtained profile.

The range of CCC lies between −1 and 1. Obtained profiles with minor shape deviations have CCC close to 1 and can be difficult to distinguish from each other. Therefore, to better distinguish such profiles, the reverse cross-correlation coefficient $CCC_{rev}$ was introduced:$$CCC_{rev}=\frac{1}{1-CCC}$$

Moreover, it is important to note that the position of the maximum of the cross-correlation function equals the position shift (in the case of harmonic textures, the phase shift) of the obtained texture relative to the position of the target texture.

#### 2.2.3. Spectral Entropy

Spectral entropy is a spectrum distribution metric. It is based on Shannon entropy in information theory. The spectral entropy SE is calculated as follows:$$SE=\frac{\sum_{n=1}^{n=f_{s/2}}S\left(n\right)log\left(S\left(n\right)\right)}{log(f_{s/2})}$$

This metric is useful for detecting harmonic textures, since the theoretical SE value of an infinite perfect sinus texture is zero, and any shape deviation leads to an increase in the spectral entropy.

### 2.3. Statistical Analysis of Metrics and Metrics Comparison

In order to compare metrics efficiency, two binary classification tests have been provided:To select the textures most similar to the target texture;To determine the presence of texture on the surface.

The following null hypotheses have been formulated:*Hypothesis 1*: the obtained texture is not similar to the target texture;*Hypothesis 2*: the obtained texture is visible on the profile.

For statistical analysis, a data set consisting of 270 profiles processed by Waveshape has been used. The profile properties are described in the Section 2.1.

Since there is no reference method for estimating the texture shape deviation, the authors adopted expert assessment of textures as the reference method. The assessment scale was as follows:3 points—good similarity: the profile is very close to the target shape, no visible shape deviations;2 points—moderate similarity: the profile is close to the target shape, minor shape deviations (regular or stochastic);1 point—weak similarity: the target texture shape is distinguishable, large shape deviations (both regular and stochastic);0 points—no texture: the target shape is not distinguishable on the cross-section.

The figures with example textures for each level of the scale are shown in Figure 6.

The scores of all the experts were summed up. Then, the assessment thresholds were set:*Hypothesis 1*: all profiles with 10 or more points were assessed as “similar”;*Hypothesis 2*: all profiles with 1 point or less were assessed as “no texture”.

The thresholds have also been set for metrics values. The threshold for each shape deviation metric has been determined in such a way that the statistical significance level (α) of the null hypothesis was no more than 5%. The specified level of statistical significance enables us to calculate the parameters of the confusion matrix.

For each metric, the parameters of the confusion matrix were predefined (Figure 7):P—the number of real positive cases in the data set;N—the number of real negative cases in the data set;TP—the number of cases that correctly indicate the presence of the effect (rejecting null hypothesis);TN—the number of cases that correctly indicates the absence of the effect (accepting null hypothesis);FP—the number of cases which wrongly indicates the presence of the effect (type I error);FN—the number of cases which wrongly indicates the absence of the effect (type II error).

Based on the confusion matrix data, the following statistical indicators can be calculated to compare metrics:True positive rate or sensitivity (TPR) means the probability of a positive result, conditioned on truly having the condition:
$$TPR=\frac{TP}{P}$$True negative rate or selectivity (TNR) means the probability of a negative result, provided one does not have the condition:
$$TNR=\frac{TN}{N}$$Positive predictive value (PPV) means the proportion of true positive results in all positive tests:
$$PPV=\frac{TP}{TP+FP}$$Negative predictive value (NPV) means the proportion of true negative results in all negative tests:
$$NPV=\frac{TN}{TN+FN}$$Accuracy (ACC)—proportion of correct results (both true positives and true negatives) in the total number of tests examined:
$$ACC=\frac{TP+TN}{P+N}$$Matthews correlation coefficient (MCC) is a measure of the binary classification quality [24]:
$$MCC=\frac{TP\times TN-FP\times FN}{\sqrt{\left(TP+FP)(TP+FN)(TN+FP)(TN+FN\right)}}$$MCC ranges from −1 to +1 where, +1 represents a perfect prediction; 0 an average random prediction; and −1 an inverse prediction.Significance level (α) means the probability of mistakenly rejecting the null hypothesis, given that the null hypothesis is true:
$$\alpha =\frac{FP}{N}.$$

## 3. Results and Discussion

The following steps have been performed to compare shape deviation metrics:Calculation of metrics thresholds for *hypothesis 1* and *hypothesis 2*;Calculation of confusion matrices for each metric and hypothesis;Calculation of statistical indicators.

The metrics have been compared by value of statistical indicators, in particular, Matthews correlation coefficient (MCC).

The significance level of the null hypothesis α=5% has been used to calculate the metrics threshold, i.e., threshold for each metric has been set at the level where the ratio of the number of of false positive results (FP) and real negative results (*N*) is equal to 0.05. The calculated thresholds for each metric and hypothesis are presented in Table 2.

The visualisation of metrics distribution with calculated thresholds according to the expert assessment are presented in Figure 8 and Figure 9.

All metrics except $R_{2/1}$ show a definite trend, increasing ($CCC_{rev}$) or decreasing ($R_{all/1}$, SE) the metric value with growth of expert assessment scores.

The values of confusion matrices for each metric and hypothesis are shown in Table 3 and Table 4.

The statistical indicators of the metrics when null *hypothesis 1* (Table 5) is rejected draw attention to the sensitivity (TPR) of all metrics with the exception of $R_{2/1}$, ranging from 0.69–0.93 with a given level of selectivity (TNR) of 0.95 (selectivity for all metrics is the same, as it is equal to 1 − α). Moreover, the positive predictive value for three metrics except $R_{2/1}$ (PPV, prediction of profile similar to the target texture) fluctuates around 0.5, the negative predictive value (NPV) is much higher and amounts to 0.98–0.99. The accuracy (ACC) of the metrics is 0.94–0.95. The Matthews correlation coefficient (MCC) is the best for spectral entropy SE and is 0.70.

The statistical indicators of the metrics, rejecting null *hypothesis 2* (Table 6), show better predictive power. The sensitivity (TPR) of all metrics, with the exception of $R_{2/1}$, are within 0.82–0.92 at a given selectivity level (TNR) of 0.95. The positive predictive value (PPV) is located within 0.66–0.68, and the negative predictive value (NPV) is 0.98–0.99. The accuracy (ACC) of the metrics is 0.94. The Matthews correlation coefficient (MCC) is also the best for spectral entropy SE and is 0.77.

The $R_{2/1}$ distribution and the values of metric statistical indicators show that this metric is poorly suited both for separating good metrics, and for finding profiles with absence of texture. The MCC score of $R_{2/1}$ is definitely lower than for the other metrics for both hypotheses I and II. For weak textures, the spectral magnitude values decrease. The random component of the first and the second harmonic peaks becomes very large, and their ratio can vary over a wide range. Thus, this reason can affect the quality of identification of textures similar to the target in case of a large number of profiles with weak textures or absence of textures.

The other metrics used have similar values for quality indicators. Only spectral entropy SE has slightly better MCC than $R_{all/1}$ and $CCC_{rev}$. The statistical indicators show that used metrics are better for identification of profiles without texture, but in general ACC and MCC show that all these metrics are applicable for determination of harmonic textures similar to the target, as well as for determination of profiles with absence of texture.

Nevertheless, the restrictions of used metrics are different.

$R_{all/1}$ is suitable for harmonic textures only, since the main harmonic is used in the calculations. SE also has a better dynamic range for harmonic textures because the spectral entropy of infinite sinus signal is equal to zero. In case of more complex target textures, SE will have the lower dynamical range and perhaps identification of target-like textures will be less successful, but this thesis requires further investigation. In general, the shape evaluation of complex textures can be performed by comparison with the target texture such as evaluation of $CCC_{rev}$.

Other constraints are imposed on the $CCC_{rev}$ metric. In cases of variability of the spatial properties, the value of $CCC_{rev}$ should be lower because of the mismatch of the obtained and the target texture. To exclude the discrepancy by evaluation of the metric, the real spatial wavelength was calculated in this study. Then, the target texture has been recalculated with the new spatial parameters. However, the spatial wavelength can vary within the texture. In this case, $CCC_{rev}$ should also decrease.

Metrics can also be sensitive to texture artifacts. Stochastic peaks on texture and texture irregularity can dramatically change the values of metrics. Texture irregularity reduces $CCC_{rev}$. A possible reason of decreasing the influence of artifacts on metric value is the use of windowing during metrics calculation. The robustness of metrics to texture artifacts should be investigated additionally.

Table 7 provides a comparison of metrics by their applicability for separating similar and non-similar textures.

## 4. Conclusions

In this study, four metrics of texture shape quality have been investigated: the magnitude ratio of the second and the first harmonics $R_{2/1}$, the ratio of the the sum of magnitudes greater than the second harmonic to the first harmonic $R_{all/1}$, the reverse value of cross-correlation coefficient $CCC_{rev}$, and the spectral entropy SE. The metrics have been analyzed on a data set of 270 texture profiles processed by laser texturing by remelting (Waveshape). The profiles have the harmonic shape, but vary widely in height and spatial dimensions: texture heights range from a few microns to several hundreds of microns, and spatial wavelengths range from 10 to 50 mm. An expert assessment of the profile shape has been provided as an independent evaluation method. Two null hypotheses have been investigated: the obtained texture is not similar to the target texture and presence of texture on the profile. The significance level of both hypotheses, which have been set on 5%, helped to evaluate the threshold levels for each metric. Binary classification analysis provided an opportunity to compare the metrics with each other by their ability to reject the null hypotheses.

According to metric distribution in dependence on expert assessment and statistical indicators (such as accuracy, Matthews correlation coefficient (MCC)), the ratio of the second and the first harmonic magnitudes $R_{2/1}$ is not applicable both for identification of textures similar to the target and for determining the presence of texture.

The other metrics have a smoother distribution and higher statistical indicators. The Matthews correlation coefficients for *hypothesis 1* (similar/not similar texture) are 0.54 for $R_{all/1}$, 0.58 for $CCC_{rev}$ and 0.66 for SE; for *hypothesis 2* (texture/no texture), they are 0.70 for $R_{all/1}$, 0.75 for $CCC_{rev}$ and 0.79 for SE. The best values of statistical indicators are obtained for spectral entropy SE. Moreover, the statistics of *hypothesis 2* show that the metrics used can distinguish presence or absence of texture better than they can pick out textures similar to harmonic shape.

Nevertheless, the used metrics have some limitations. Theoretically, the ratio of the the sum of magnitudes greater than the second harmonic to the first harmonic $R_{all/1}$ works only in the case of harmonic textures, where one main wave is significantly larger than other. This statement may also be true for spectral entropy SE but requires proof. The reverse value of cross-correlation coefficient $CCC_{rev}$ has no constraints on the texture shape, since the metric is the result of comparing target and obtained texture. However, the irregularity of texture period can rapidly increase the inverse cross-correlation coefficient $CCC_{rev}$.

Metric analysis has been implemented on textures created by Waveshape. However, there are no limitations associated with the texturing method. In particular, textures created by other laser texturing methods, such as LIPSS, DLIP, LST, are in most cases regular and periodic. The metrics used, especially $CCC_{rev}$, can be helpful for determining the shape deviation of such textures.

Further, it is planned to expand the use of metrics for 3D textures, as well as non-harmonic textures, to study the robustness of metrics to artifacts of processing with windowing of textures, and to develop a method for searching laser texturing parameters using the metrics presented in this work as processing quality characteristics.

## Figures and Tables

**Figure 1 micromachines-13-00618-f001:**
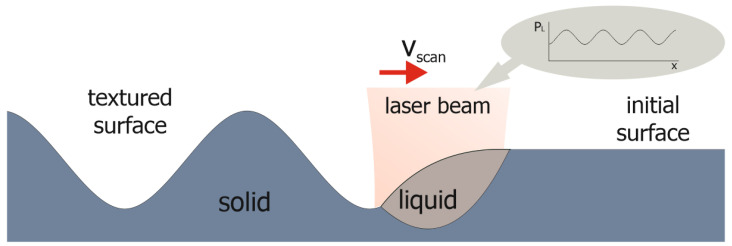
Scheme of Waveshape process.

**Figure 2 micromachines-13-00618-f002:**
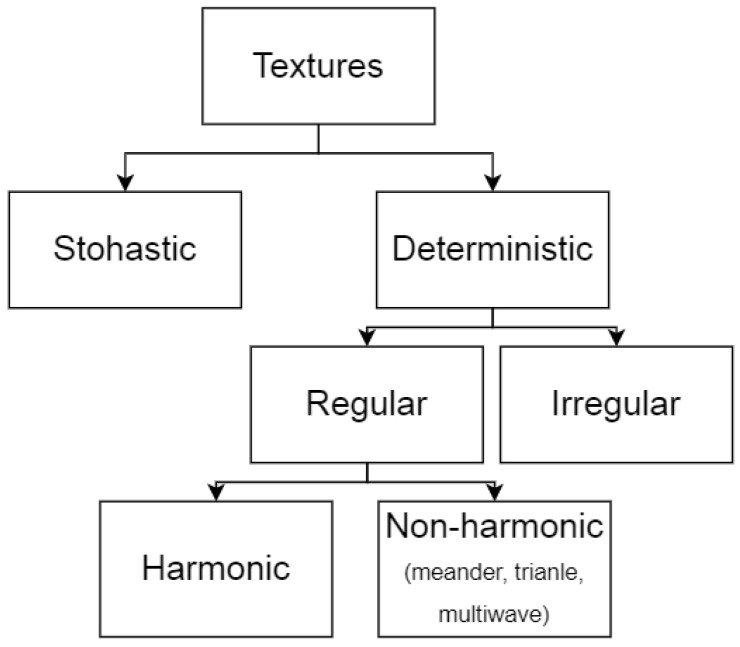
Types of surface texture patterns.

**Figure 3 micromachines-13-00618-f003:**
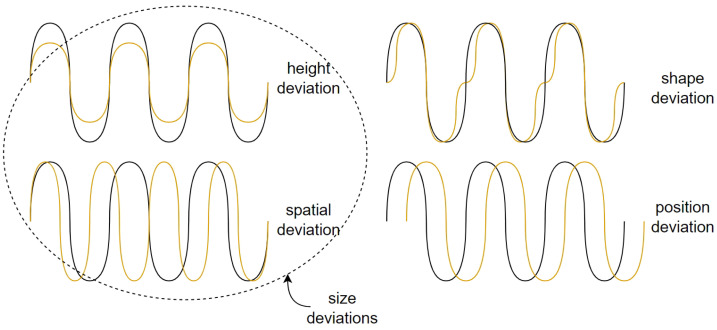
Types of texture deviations.

**Figure 4 micromachines-13-00618-f004:**
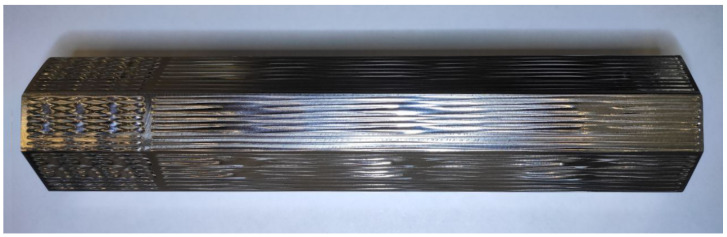
Octagonal Ti-6Al-4V alloy sample with textured single lines on the surface.

**Figure 5 micromachines-13-00618-f005:**
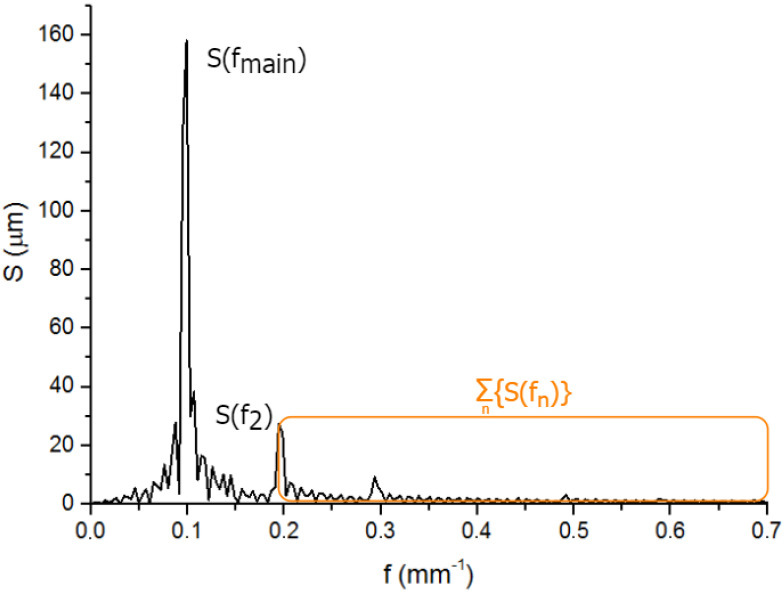
The components of spectral metrics labeled by the example of the profile spectrum.

**Figure 6 micromachines-13-00618-f006:**
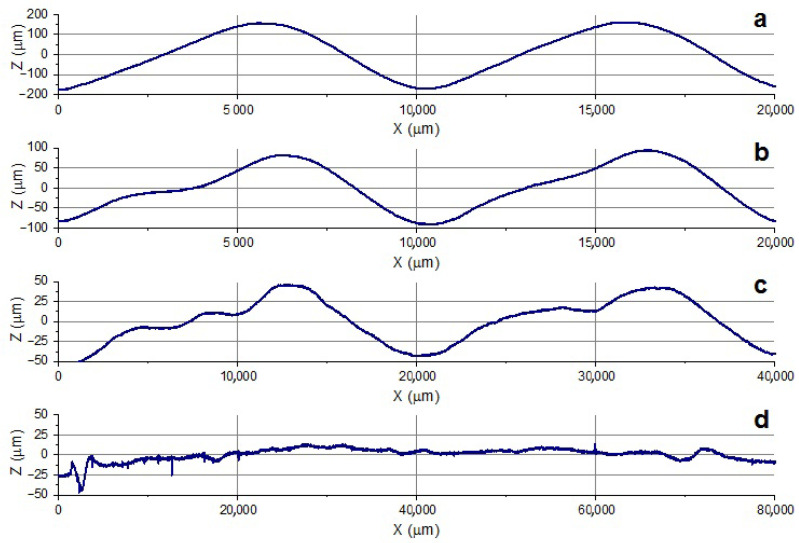
Examples of profiles with the textures obtained by Waveshape: (**a**) good similarity to the target texture, (**b**) moderate similarity to the target texture, (**c**) weak similarity to the target texture, (**d**) texture absence.

**Figure 7 micromachines-13-00618-f007:**
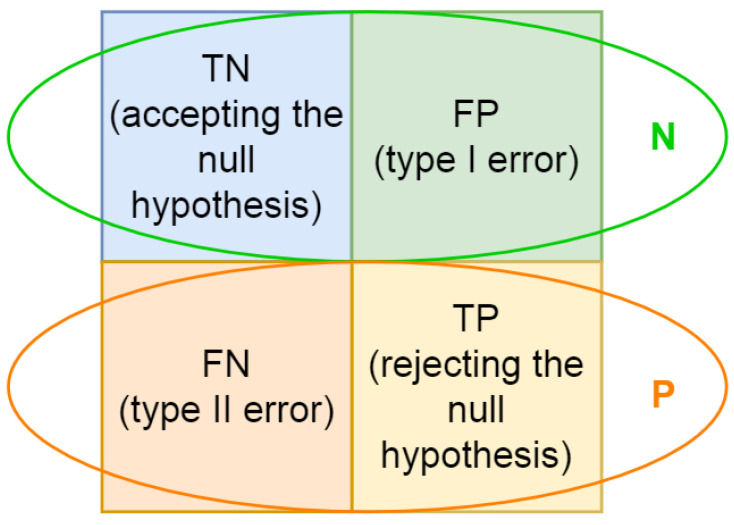
Confusion matrix for binary classification.

**Figure 8 micromachines-13-00618-f008:**
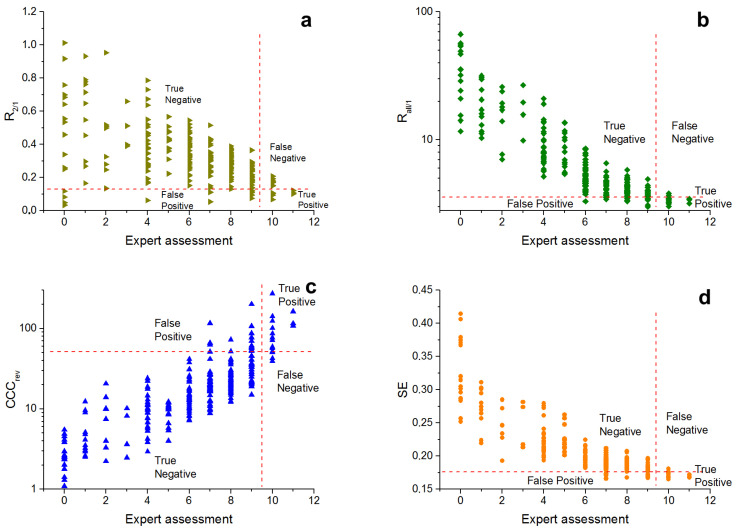
Distribution of the texture metrics in accordance to the expert assessments: $R_{2/1}$ (**a**), $R_{all/1}$ (**b**), $CCC_{rev}$ (**c**), SE (**d**). Thresholds are given for *hypothesis 1*.

**Figure 9 micromachines-13-00618-f009:**
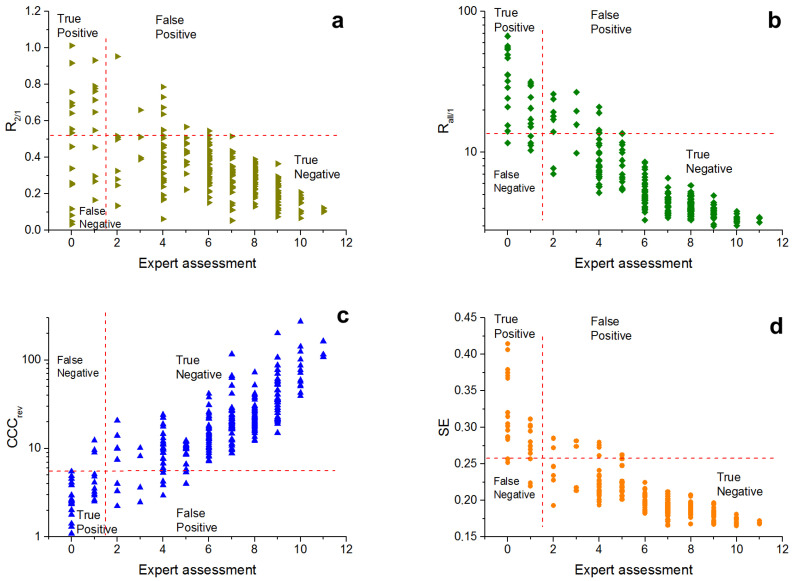
Distribution of the texture metrics in accordance to the expert assessments: $R_{2/1}$ (**a**), $R_{all/1}$ (**b**), $CCC_{rev}$ (**c**), SE (**d**). Thresholds are given for *hypothesis 2*.

**Table 1 micromachines-13-00618-t001:** Waveshape processing parameters used to obtain the profiles data set.

Parameter	Designation	Value
Mean laser power range [W]	$P_{M}$	385–1375
Laser power modulation range [W]	$P_{M}$	176–484
Variation of spatial wavelength [mm]	λ	10, 20, 30, 40, 50
Variation of scan speed [mm/s]	$v_{scan}$	25, 50, 100
Variation of laser beam diameter [μm]	$d_{L}$	500, 750, 1000
Number of processing repetitions	*n*	1, 2, 4, 8, 16, 32
Processing gas	-	Ar
Residual oxygen concentration [ppm]	-	100

**Table 2 micromachines-13-00618-t002:** Threshold values of metrics for binary classification of *hypothesis 1* and *hypothesis 2*.

Parameter	*R* _2/1_	*R* _*all*/1_	*CCC* _ *rev* _	*SE*
*Hypothesis 1*	0.12	3.47	53.95	0.177
*Hypothesis 2*	0.52	13.93	5.77	0.246

**Table 3 micromachines-13-00618-t003:** Parameters of confusion matrices for the *hypothesis 1*.

Parameter	*R* _2/1_	*R* _*all*/1_	*CCC* _ *rev* _	*SE*
TP	8	11	12	15
FP	12	12	12	12
TN	242	242	242	242
FN	8	5	4	1

**Table 4 micromachines-13-00618-t004:** Parameters of confusion matrices for *hypothesis 2*.

Parameter	*R* _2/1_	*R* _*all*/1_	*CCC* _ *rev* _	*SE*
TP	16	23	25	26
FP	12	12	12	12
TN	230	230	230	230
FN	12	5	3	2

**Table 5 micromachines-13-00618-t005:** Metric statistical indicators for *hypothesis 1*.

Indicator	*R* _2/1_	*R* _*all*/1_	*CCC* _ *rev* _	*SE*
TPR	0.50	0.69	0.75	0.93
TNR	0.95	0.95	0.95	0.95
PPV	0.40	0.48	0.50	0.55
NPV	0.97	0.98	0.98	0.99
ACC	0.93	0.94	0.94	0.95
MCC	0.41	0.54	0.58	0.70

**Table 6 micromachines-13-00618-t006:** Metric statistical indicators for *hypothesis 2*.

Indicator	*R* _2/1_	*R* _*all*/1_	*CCC* _ *rev* _	*SE*
TPR	0.57	0.82	0.89	0.92
TNR	0.95	0.95	0.95	0.95
PPV	0.57	0.66	0.68	0.68
NPV	0.95	0.98	0.99	0.99
ACC	0.91	0.94	0.94	0.94
MCC	0.52	0.70	0.75	0.77

**Table 7 micromachines-13-00618-t007:** Applicability of shape deviation metrics on identification of target-like textures.

Parameter	*R* _2/1_	*R* _*all*/1_	*CCC* _ *rev* _	*SE*
Type of texture	harmonic	harmonic	any	any (possible non-harmoninc worse)
Deviation of texture period	Low influence	Low influence	Large influence	Low influence

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
