# Peer review of "Comparison of Metrics for Shape Quality Evaluation of Textures Produced by Laser Structuring by Remelting (Waveshape)"

_micromachines, 2022, doi:10.3390/mi13040618_

Round 1

Reviewer 1 Report

The study proposes four metrics and their calculation methods to evaluate the surface texture geometrical quality and has included enough considerations in the method design. Furthermore, the work also compared restrictions of different methods and discussed suitabilities. This work, I think, is an endeavour work worthing spreading, is meaningful for the engineering field.

I suggest a Minor Revision before accepting it to be published in Micromachines.

Here are some questions/ suggestions from me.

  1. As it is an interesting and meaningful topic for many fields, I suggest briefly introducing the significance of this research in the Abstract.
  2. The four metrics are all from the statistical calculation. It is good to compare the four metrics so we can know the difference and limitations among the four. However, is there any validation work done to verify the calculated results. The validation is, for example, to compare the calculated data with the actual texture shapes produced by the laser process.
  3. After the laser process and before geometrical measurements, is there any procedure to check the texture processing quality (like metallurgical cracks, pores, etc.)? Is every surface texture cell having good quality for the next geometrical metric calculation?
  4. Suggest providing the target size you were going to fabricate before the laser process, and then the obtained size after the laser process, so authors can compare their own work with your results. Also, it is preferable to provide some typical optical images or SEM images of the obtained surface textures.
  5. A small grammar error on Line 65: Deterministic textures can also ME measured by position.

Author Response

Dear Reviewer,

thank you very much for your comments.

Our response is attached.

Best Regards

Oleg Oreshkin

Reviewer 2 Report

Everything is fine

Author Response

Dear Madam or Sir,

thank you very much for reviewing our manuscript and appreciation of our work.

Best Regards,

Oleg Oreshkin